# Comparison of Polypropylene and Ceramic Microfiltration Membranes Applied for Separation of 1,3-PD Fermentation Broths and *Saccharomyces cerevisiae* Yeast Suspensions

**DOI:** 10.3390/membranes11010044

**Published:** 2021-01-08

**Authors:** Wirginia Tomczak, Marek Gryta

**Affiliations:** Faculty of Chemical Technology and Engineering, West Pomeranian University of Technology in Szczecin, ul. Pułaskiego 10, 70-322 Szczecin, Poland

**Keywords:** 1,3-propanediol, ceramic membrane, clarification, fermentation broth, fouling, membrane cleaning, membrane wetting, microfiltration, polymeric membrane, yeast suspension

## Abstract

In recent years, microfiltration (MF) has gained great interest as an excellent technique for clarification of biological suspensions. This paper addresses a direct comparison of efficiency, performance and susceptibility to cleaning of the ceramic and polymeric MF membranes applied for purification of 1,3-propanediol (1,3-PD) fermentation broths and suspensions of yeast *Saccharomyces cerevisiae*. For this purpose, ceramic, titanium dioxide (TiO_2_) based membranes and polypropylene (PP) membranes were used. It has been found that both TiO_2_ and PP membranes provide sterile permeate during filtration of 1,3-PD broths. However, the ceramic membrane, due to the smaller pore diameter, allowed obtaining a better quality permeate. All the membranes used were highly susceptible to fouling with the components of the clarified broths and yeast suspensions. The significant impact of the feed flow velocity and fermentation broth composition on the relative permeate flux has been demonstrated. Suitable cleaning agents with selected concentration and duration of action effectively cleaned the ceramic membrane. In turn, the use of aggressive cleaning solutions led to degradation of the PP membranes matrix. Findings of this study add to a growing body of literature on the use of ceramic and polypropylene MF membranes for the clarification of biological suspensions.

## 1. Introduction

Microfiltration (MF) is one of the oldest membrane technologies [1], which is characterized by operating pressure lower than 0.35 MPa and high permeate fluxes, mainly between 10^−4^ and 10^−2^ m/s [2]. Over the last two decades, many attempts have been made by researchers to comprehensively investigate the use of MF in the clarification of fermentation broths [3,4,5,6,7,8,9,10,11,12,13,14,15,16,17,18] and yeast suspensions [19,20,21,22,23,24,25,26,27,28,29,30,31,32]. It is due to the fact that separation of biological materials using conventional centrifugal methods is difficult and expensive [33]. In turn, MF is an effective technique for turbidity removal [34] and offers several remarkable advantages, such as low cost, easy operation and implementation, high productivity, absence of phase transition, non-use of additional solvents and so on [8].

Nowadays, advanced clarification of biological suspensions can be performed using a wide variety of commercially available MF membranes. Moreover, according to [35] the global market of MF membranes should increase from $2.4 billion in 2018 to $3.7 billion by 2023. Nevertheless, predicting the most suitable membrane for the identified application is a challenging task and requires a multi-criteria approach. Speed [36] has indicated that among the most important factors in selecting the right membrane are mechanical strength, resistance to cleaning chemicals, pore size and surface charge. Additionally, since the decrease in permeate flux is one of the most important economic criterions for a given separation process, due attention should be paid to the membrane performance. It is well established that the fouling phenomenon is a complex physicochemical phenomenon that depends on the process operating parameters and the interactions between the membrane and feed components. Moreover, it leads to the necessity of frequent membrane cleaning. Consequently, the extensive knowledge of the membrane material and specification is a key factor in successfully realizing an efficient and economically viable MF process. Moreover, as it has been pointed out by Warsinger et al. [37], comparing different types of membrane materials leads to an understanding of their advantages and disadvantages.

In recent years, research on the separation processes using ceramic membranes has been receiving growing attention. Indeed, Li et al. [38] in the newly published review article have demonstrated that the publications focused on ceramic membranes have been expanding over the last 10 years, from less than 150 in 2007 to double that in 2018. It is due to the distinguished merits of ceramic membranes over their polymeric counterparts. As recognized in the literature [24,39,40,41,42,43,44], ceramic membranes offer mechanical strength and resistance to harsh chemical conditions. In addition, they can withstand temperatures of up to 500 °C [24]. The above-mentioned advantages of ceramic membranes enable their specialized use in extreme operating conditions. For instance, they allow for aggressive physical and chemical cleaning of the modules, which ensures the removal of irreversible fouling, without the risk of damaging membrane integrity. Moreover, ceramic membranes show excellent corrosion resistance [41] as well as inertness to microorganisms [24] and organic media [45]. The high reliability of ceramic membranes decreases the cleaning requirements, reduces the replacement of membrane modules and, thus, prolongs their operational life expectancy. Nevertheless, ceramic membranes are more prone to breakage than polymeric membranes [46], thus, they need to be handled carefully.

Nowadays, ceramic membranes are available in different configurations, among which tubular-shaped clearly dominate the field [47]. Generally, ceramic membranes have a multilayer structure consisting of one or more different inorganic materials. In industry, the composition of ceramic membranes is usually based on alumina (Al_2_O_3_), zirconia (ZrO_2_), titania (TiO_2_) or combination of these oxides [47] and their characteristics depend on the materials involved. Since the surface groups of ceramic and polymeric membranes are different, differences in their fouling are expected [46]. Indeed, ceramic membranes are generally hydrophilic in nature which ensures a lower protein adhesion than hydrophobic membranes, such as polypropylene and polyethylene [42]. Moreover, they have relatively narrow pore size distribution and higher porosity, resulting in better separation characteristics and a higher flux [46].

It is well recognized that the high prices of raw materials and manufacturing are identified as the main disadvantages of ceramic membranes. Indeed, Issaoui and Limousy [44] have noted that the price of an α-alumina porous tubular ceramic membrane with average pore diameters in the range from 1000 to 6000 nm ranges between $500 and $1000/m^2^. Definitely, the cost of ceramic membranes is the main limiting factor in their large-scale industrial applications. In light of this, polymeric membranes can be considered an alternative.

The most commonly used polymers in membrane production are polypropylene (PP), polyvinylidenefluoride (PVDF), polyacrylonitrile (PAN), polyethersulfone (PES), cellulose acetate (CA), polyamide (PI) and polyvinylpyrrolidone (PVP) [48]. Among the main advantages of polymeric membranes are the low cost of production and possibility to synthesize novel polymers with well-defined structures [49]. In addition, they show minimal interaction with organic compounds [50]. However, due to their nature, they are characterized by a reduced chemical stability to high and low pH, which limits the possibilities of the chemical cleaning used [51]. It is an important issue which must be considered, since for cleaning membranes fouled by bacterial suspensions, the most commonly used are aggressive cleaning agents, such as: sodium hydroxide (NaOH) and sodium hypochlorite (NaOCl) [5]. Indeed, it has been widely reported [52,53,54,55,56,57,58,59] that chemical cleaning of polymeric membranes may lead to modification of their hydraulic performances, mechanical properties and physical structures. For instance, Malczewska and Żak [54] have investigated the impact of NaOH, hydrochloric acid (HCl) and NaOCl solutions on the properties of a flat sheet PES ultrafiltration membrane. The authors have demonstrated that NaOCl led to degradation of the membrane most quickly. In addition, they found changes in the membrane surface properties, morphology and hydraulic performance caused by NaOCl, indicating that it could cause ageing of the membrane after a prolonged exposure. In turn, in [56] it has been shown that NaOCl may lead to ageing on the PVDF membrane after prolonged exposure and significant changes in its mechanical properties, hydraulic performance and chemical groups as well as physical structures. 

As stated before, the membrane material is considered as one of the major factors playing a role on the membrane fouling phenomenon. Hence, in the last years, an important research effort has been made on the comparison of performance of ceramic and polymeric membranes used in various pressure driven separation processes, such as: MF of milk [51], surface water [46] and algal rich water [60], MF and ultrafiltration (UF) of ground water [61] and real produced water [62], UF of synthetic feedwater [63,64], oil-in-water emulsion [65] and cadmium(II) ions from aqueous solutions [49] as well as nanofiltration (NF) of trace organic chemicals [66] and acid mine waters [67]. For instance, Vatai et al. [65] have demonstrated the comparison of a ceramic ZrO_2_ membrane (nominal pore size of 20 nm) with a polyaryletherketone (PAEK) membrane (MWCO of 100 kDa) applied for UF of oil-in-water emulsion. The authors have pointed out that, although oil rejection of a ceramic membrane was not satisfactory, its advantage over a polymer membrane in terms of productivity has been found. Moreover, the necessity of conducting the filtration process under optimal operating conditions, especially in the case of ceramic membranes, was emphasized. Also, the authors have highlighted their better mechanical, thermal and chemical stability. In turn, Kenari et al. [63] have investigated the UF process of synthetic water with using two types of membranes: a ceramic tubular membrane made of ZrO_2_ and a support of *α*-Al_2_O_3_ (MWCO of 150 kDa) and a polymeric hollow fiber membrane composed of a blend of PVP and PES (MWCO of 200 kDa). The authors have shown that, although both membranes used were similarly fouled during the investigation process, the physical cleaning was more effective for a ceramic membrane than a polymeric one.

It is essential to mention that in the available literature studies comparing the performance of polymeric and ceramic membranes used for clarification of biological suspensions have not been found. Therefore, much effort should be devoted to comparing performances of ceramic and polymeric membranes applied for MF of 1,3-propanediol (1,3-PD) fermentation broths and suspensions of yeast *Saccharomyces cerevisiae*. The motivation lies in the fact that 1,3-PD is one of the most important organic chemical materials which is being extensively applied in a range of industrial applications. For instance, it is used for synthesis of a biodegradable polymer known as polytrimethylene terephthalate (PTT) as well as production of detergents, cosmetics, composites and lubricants [23,68,69,70,71]. However, fermentation broths of 1,3-PD are complex media which, apart from the desired product, contain water, microbial cells, residual carbon source (e.g., glycerol) and various by-products (e.g., acetic acid, lactic acid, ethanol and 2,3-butanediol) [72]. Hence, in order to obtain the pure main product, the clarification of post-fermentation solution as the first step of downstream processing is required. In subsequent stages, metabolites are separated by the nanofiltration (NF) and reverse osmosis (RO) [73]. The spiral wound structure of membrane modules used in these processes requires careful removal of turbidity (Silt Density Index < 5) and sterility of the obtained MF permeate. 

In turn, interest in the yeast filtration is related to the fact that *Saccharomyces cerevisiae* is the most intensively studied unicellular eukaryote and one of the main industrial microorganisms [74]. Indeed, its cytoplasm is a rich source of various bioproducts (e.g., proteins, polysaccharides and cytoplasmic enzymes) that are valuable in a massive variety of fields, such as the beverage and food industry, biotechnology and pharmacology [23]. Furthermore, *Saccharomyces cerevisiae* is one of the hosts for synthetic biology [20]. Importantly, this microorganism has well-defined granulometric properties, its suspension is easily prepared feed and, hence, it is an excellent model system for studying the behavior and treatment of biological materials [21,24].

Given the background described above, this work addresses the direct comparison of efficiency, performance and susceptibility to cleaning of the ceramic and polymeric microfiltration membranes applied for clarification of 1,3-propanediol fermentation broths and aqueous suspensions of dry distillery yeast *Saccharomyces cerevisiae*-Bc 16a. For this purpose, the membranes made of TiO_2_ and PP in different module configurations (tubular and capillary) were tested. 

## 2. Materials and Methods

### 2.1. Feed Suspensions

The glycerol fermentation was carried out in a LiFlusGX bioreactor (Biotron Inc., Korea), using *Citrobacter freundii* and *Lactobacillus casei* bacteria isolated and characterized in Department of Biotechnology and Food Microbiology, Poznań University of Life Science (Poland). The process has been described in detail in our previous works [72,75]. The obtained 1,3-PD post-fermentation solutions were subsequently used as feed in the studied MF processes. Their compositions are presented in Table 1.

Suspensions of commercially available distillery yeast *Saccharomyces cerevisiae*-Bc 16a (Wytwórnia Drożdży Maszewo Lęborskie, Poland) were also used as feed in this study. The yeast concentrations of 0.1, 0.16 and 0.5 g/L were prepared by mixing dry yeast cells and distilled water at room temperature for 30 min.

### 2.2. Microfiltration Set-Up and Process Parameters

Microfiltration experiments were carried out in the installations (Figure 1), the schemes of which were described in detail in our previous studies [5,76].

In the present research, four different modules with MF membranes were used (Table 2). Modules M1 and M2, without an external shell, were tested in the installation shown in Figure 1a. In each of these modules, the four capillary membranes made of polypropylene were assembled. In turn, in the pilot installation presented in Figure 1b, tubular modules (diameter 12 mm) with the polypropylene membrane (module M3) or ceramic membrane (M4) were tested.

Due to the hydrophobic properties of polypropylene, water does not wet PP membranes, while water penetration into the pores (Liquid Enter Pressure—LEP) occurs under the pressure of 0.2–0.3 MPa. Therefore, prior to the MF experiments, a membrane wetting operation was required. For this purpose, the Accurel PP S6/2 membranes assembled in the modules M1 and M2 were submerged into ethanol for 15 min, and then the filtration of ethanol (100 mL) was conducted. Subsequently, in order to remove residual ethanol, the membranes were flushed by distilled water (2 L) and the permeate flux J as a function of transmembrane pressure (TMP) was determined. Then, the filtration of distilled water under TMP equal to 0.03 MPa for 4 h was performed. Due to the instability of the recorded permeate flux values, the next day water was removed from the system, the membranes were immersed in ethanol and the filtration was conducted for 15 min. Then, alcohol was rinsed with water and the relationship J = f(TMP) was studied. Subsequently, water filtration was continued under TMP of 0.03 MPa for 2 h. Then, water was replaced with ethanol and the above-described operation was repeated two times. Since the obtained relationship between the permeate flux and TMP was repeatable, the studies on MF of biological suspensions were started. In turn, the Accurel PP V8/2 HF membrane assembled in the M3 module was wetted by filling the outer space of the module with ethanol and filtering it for 15 min. Subsequently, the water filtration was carried out for 30 min. In the next step, the water flux as a function of TMP was determined using a pressure ranging from 0.04 to 0.1 MPa. Finally, the water filtration was continued for several hours, obtaining a constant relationship J = f(TMP), independent of the feed flow rate.

Microfiltration experiments of 1,3-propanediol fermentation broths and yeast *Saccharomyces cerevisiae* suspensions have been carried out at constant temperature of 30 °C and across a wide range of process parameters. The TMP ranging from 0.02 to 0.08 MPa was used. In turn, the applied feed flow velocity V_F_ was from 0.8 to 11 m/s.

The permeate flux J was calculated based on the measured volume of collected permeate according to Equation (1), as follows:(1)$$J=\frac{\Delta V}{S\Delta t}$$
where ∆V is the permeate volume (L) collected over Δt period (h), S is the total active membrane area (m^2^). 

Fouling intensity was determined by measuring the relative permeate flux, defined as the ratio between the actual permeate flux J and maximum permeate flux J_0_ for a clean membrane.

Cleaning efficiency was estimated from the ratio of water permeate fluxes after membrane cleaning to the maximum flux, obtained under the same operational conditions (temperature, transmembrane pressure and feed flow velocity).

### 2.3. Analytical Methods

Permeate and feed samples were analyzed in terms of compounds content, turbidity and number of bacteria. The analytical methods used for this purpose were described in detail in our previous work [5]. The morphology of the membrane surface was examined by scanning electron microscopy (SEM) (SU-70 and SU-8000, Hitachi High Technologies Co., Tokyo, Japan).

## 3. Results

### 3.1. Membranes Morphology and Maximum Performance

The polypropylene membranes used for the present studies were formed via TIPS method. It has been determined that, due to the interaction with the gelling bath, the porosity of the membrane surface (Figure 2a) slightly differs from that observed inside the wall (Figure 2b). In turn, the ceramic membrane manufactured by Tami Ind. had both a support layer and a thin active layer made of TiO_2_ (Table 2). The structure of these layers was crystalline, ranging from a small size in the surface active layer (Figure 2c) to over 20 µm in the support layer (Figure 2d). 

As reported in Section 2.2, an operation of wetting the PP membranes with ethanol was carried out prior to the MF experiments. Subsequently, in order to remove the solvent, the membranes were shortly rinsed with distilled water. However, the performed operation was not effective since the membranes regained their hydrophobic properties. Probably, the new PP membrane, after washing the ethanol with water, shows a strong tendency to push the water out of the pores and part of its volume is again filled by air, which blocks the flow of water through the pores. Indeed, after 2–3 h of the water filtration a rapid decline in the permeate flux was observed. For instance, for TMP equal to 0.05 MPa, a decrease in the M2 module performance from 630 to 112 L/m^2^h was reported. 

An important point which should be noted is that stable modules performance (Figure 3, lines L2) was obtained after 2–3 days of the rinsing, during which the operation of membrane wetting was repeated several times. Moreover, the results presented in Figure 3 show that the PP membranes performance after wetting depends on the pumping pressure of the rinse water. The M3 module (V8/2 HF membrane) was rinsed under much higher pressure (0.04–0.1 MPa) than the S6/2 membranes assembled in M1 and M2 modules (0.01–0.05 MPa). Probably, the use of higher TMP values prevents water from being pushed out of the pores and re-creating the gas phase in them, hence, the performance drop of the M3 module was much lower.

In turn, the performance of the ceramic membrane used was more stable. The membrane was conditioned by rinsing with distilled water for two days, with fluctuations of the permeate flux up to 30%. Finally, the performance of the M4 module stabilized after washing the membrane with 3% NaOH solution for 60 min, and then for the same period of time with 3% H_3_PO_4_ solution. It is worth noting that under the same operating conditions, the maximum permeate flux noted for the ceramic membrane (M4) is lower than that for the V8/2 membranes (M3) (Figure 3). This observation is related to the fact that the V8/2 membranes have a larger pore size (0.20 µm) than the ceramic membrane (0.14 µm) (Table 2). Obviously, membranes with larger pore size are more porous, which means they provide a higher permeate flux [31].

Figure 3 shows the stabilized performances obtained for the tested modules fed with distilled water. Unfortunately, the permeate flux decreases significantly during filtration of real solutions. Importantly, the intensity of the flux decline, apart from the properties of membranes (hydrophilic or hydrophobic), is significantly influenced by the composition of the feed. For this reason, in order to investigate the intensity of the fouling phenomenon, in subsequent stages of the research, in addition to yeast suspension, real broths with different compositions (Table 1) were used. 

### 3.2. Filtration of Fermentation Broths with Citrobacer freundii Bacteria

The glycerol fermentation process was carried out for 48 h. Once the process run was complete, the obtained 1,3-propanediol post-fermentation solution was clarified using the MF membranes. Figure 4 shows the changes in the relative permeate flux of the M1 module during the filtration of solution with *Citrobacter freundii* bacteria. The process under TMP equal to 0.03 MPa and V_F_ of 0.8 m/s was conducted. It has been demonstrated that the permeate flux systematically decreased over 120 min. Obviously, it was caused by the membrane fouling, defined as “the accumulation of substances on the membrane surface and/or within the membrane pores, which results in deterioration of membrane performance” [77]. Subsequently, the solution was drained from the tank and the membrane module was flushed with 2 L of distilled water (TMP = 0). Then, the performance of the membrane for distilled water was controlled. It has been reported that the permeate flux during the MF of distilled water increased slightly, indicating that the presented method of the membrane cleaning was not efficient. Then, the MF process of the broth was resumed. It can be clearly observed that after three series of filtration, followed by systematically carried out water filtration, the maximum performance of the membrane module decreased by nearly 50%. 

One membrane from the M1 module was collected and the surface morphology of the selected membrane was examined using scanning electron microscopy (SEM). Meanwhile, three membranes remaining in the M1 module were rinsed with 1% NaOH solution for 30 min. It is important to note that this operation allowed to almost completely recover the maximum permeate flux (Figure 4, CR point). The high effectiveness of NaOH solution is related to the fact that at caustic conditions, large organic particles (e.g., colloids and microbes) can be disintegrated into fine particles and soluble organic matters, while organic matters (e.g., proteins and carbohydrates) can be hydrolyzed and solubilized into small molecules [78]. It is worth noting that in the study [10], NaOH solution was successfully used as a cleaning agent for microfiltration membranes made of CA and PVDF and a PES ultrafiltration membrane fouled with *Bacillus thuringiensis* fermentation broth.

Considering the application of NF/RO processes for MF permeate separation, the sterility and turbidity of filtered fermentation broths control the efficiency of the MF process. Therefore, in the present study these quality attributes of the obtained permeate were continuously controlled. It has been found that both the ceramic and PP membranes used provided a sterile permeate. Indeed, although the number of bacteria in the feed was of an order of 12 log CFU (colony-forming units-CFU/mL), no bacteria was detected in the permeate samples. 

Regarding turbidity, during the MF process with the M1 module, the broth was recirculated through the system, which resulted in a continuous concentration of the feed and, consequently, an increase in its final turbidity to about 8000 NTU (Figure 5). Despite such a high value, the turbidity-free filtrate was obtained. It is worth noting that during the first series of measurements with the new membranes, the permeate turbidity increased to 3 NTU, while in the third series it stabilized at 1 NTU. The observed decrease of the NTU values during the MF process indicated the phenomenon of membranes fouling, which was confirmed by the observed decrease in module performance (Figure 4). Roughly speaking, the membrane efficiency can be increased due to the cake formation, as particles with diameters smaller than the pore size of the membrane are more susceptible to being captured by the cake [79]. An apparent decrease in the turbidity of permeate samples during the pressure driven separation processes of fermentation broths has also been reported in several previous studies [5,7,72,80].

Figure 6 shows the SEM image of the PP S6/2 membrane surface after the MF process of fermentation broth with *Citrobacter freundii* performed under TMP of 0.03 MPa and V_F_ of 0.8 m/s. It can be clearly observed that the surface of the membrane tested was completely covered by deposit in which the *Citrobacter freundii* cells can be seen. As it has been described in [81] they are rod shaped and typically range from 1 to 5 μm in length.

It is well known that the adverse effects caused by fouling may be reduced by operating the filtration process under an increased shear rate near the membrane. Generally, enhanced hydrodynamic conditions are applied in the systems equipped with ceramic membranes. Indeed, the increase of the permeate flux with increase of the shear rate has been reported in several previous works [5,7,8,27,80,82,83,84] where the use of ceramic MF and UF membranes in clarification of various biological suspensions has been studied. A similar pattern of results was obtained in the present work. In our research, the effect of the feed velocity during the MF of 1,3-PD fermentation broth with using the ceramic membrane (module M4) under TMP equal to 0.08 MPa has been investigated. It was found that increasing the feed flow velocity from 5.5 to 11 m/s allowed the increase of the steady state permeate flux from 180 to 228 L/m^2^h. It clearly indicates that two-fold increase of the feed velocity led to an increase of the relative flux from 0.3 to 0.38 (Figure 7). This observation can be explained by the fact that the higher feed crossflow velocity led to a reduction of the deposited particles and consequently an increase in the permeate flux. However, it should be mentioned that in industry, the use of high crossflow velocities leads to high pressure losses and increased process costs [51].

Figure 8 shows the changes in the feed and permeate turbidity during the MF of 1,3-PD fermentation broth with *Citrobacter freundii* using the ceramic membrane. 

It has been found that, likewise to the process with using the PP membranes, the obtained permeate was sterile and its turbidity decreased over the MF time. It has been demonstrated that the ceramic membrane used provided obtaining the permeate characterized by the turbidity equal to 0.2 NTU at the end of the trial. The most conspicuous observation to emerge from the data comparison was the quality of permeate obtained during the MF with ceramic and polypropylene membranes. Indeed, the value mentioned above is significantly lower than that recorded during the MF process with the PP membranes (Figure 5). It is due to the fact that, as presented in Table 2, the ceramic membrane used had a nominal pore size (0.14 µm), smaller than the PP membranes (0.20 µm). These results offer vital evidence that the permeate quality is directly correlated to the membrane pore size. The similar relation between diameters of the MF membranes pore size and filtrate quality has been found in previous studies [79,85]. For instance, Mora et al. [85] investigated the MF process of a grape marc extract characterized by the turbidity equal to 640.44 ± 380.32 NTU. The authors have shown that the use of ceramic membranes with the pore sizes of 0.14 and 0.80 µm provided the permeate with 3.8 NTU and 4.8 NTU, respectively. On the other hand, our results differ from those of Ciu et al. [86] who studied the filtration process of seawater collected from a seawater desalination reverse osmosis plant. For this purpose, the authors have used ceramic membranes with pore sizes equal to 50, 200, 500 and 800 nm. They have indicated that the pore size of membranes has insignificant effects on the obtained permeate turbidity. 

Moreover, results obtained in the present study (module M4) have shown that the turbidity of the permeate was independent of the feed flow velocity (Figure 8), indicating that the separation improvement was mainly due to internal membrane fouling.

Membrane chemical cleaning is considered the most effective method to recover the permeate flux. However, its efficacy depends on several factors, such as: the membrane and fouling type, choice of cleaning agents and their concentration as well as cleaning conditions (e.g., temperature, pressure, cross-flow velocity, pH, time) [78]. However, as stated before, prolonged exposure to chemical cleaning agents may lead to physical and chemical degradation of polymeric membranes, resulting in a change in their performance and selectivity. Therefore, in order to investigate the impact of multiple cleaning cycles on the polypropylene membranes’ properties, the long-term MF process of 1,3-PD fermentation broth with *Citrobacter freundii* with periodically rinsing the module with 1% NaOH solution has been conducted (Figure 9). 

The clarification process under TMP of 0.03 MPa and V_F_ equal to 0.8 m/s has been conducted. Results obtained in our research revealed that cleaning allowed the recovery of the initial permeate flux in 60–80%. Importantly, the performance of MF with distilled water, conducted in the next step, allowed the complete recovery of the maximum permeability of the membranes. This noteworthy result suggests that the NaOH solution did not completely remove the deposit but allowed a part of it to be loosened. Hence, the water filtration provided to cleanse the deposit, especially, that accumulated in the membrane’s pores.

It is necessary to mention that, although the multiple cleaning cycles have been performed, the recorded permeate turbidity for the last two measurement series (Figure 9) was lower than 1 NTU (Figure 10). What becomes apparent from the discussed results is that in the end of the process a better separation efficiency was obtained (Figure 5).

The SEM analysis showed slight surface degradation of the PP membranes (Figure 11). Since maintaining the good separation properties of membranes has been documented (Figure 10), it can be concluded that the cleaning with NaOH solution changed the top layer of membranes, while the pores were not significantly enlarged. However, in comparison with Figure 2a, an increase in the porosity of the membrane surface can be observed. It could facilitate the penetration of the foulants deeper into the membrane wall, which has resulted in an increase in the decline of relative flux observed for the last two measurements series, shown in Figure 9.

In regards to the ceramic membrane, it has been reported that its cleaning after the filtration of 1,3-PD fermentation broths with *Citrobacter freundii* (Figure 7, TMP = 0.08 MPa, V_F_ = 5.5 m/s) with only 1% NaOH did not provide satisfactory results. Indeed, it has been found that recovering the initial performance of the module required a more complex cleaning procedure (Figure 12). In order to determine the minimum cleaning time required to recover the maximum permeate flux, cycles of 5–10 min were performed.

It has been demonstrated that the membrane rinsing with distilled water for 30 min allowed to recover the maximum membrane permeability in 40% (Figure 12). Then, in order to increase the cleaning efficiency, 3% NaOH solution was used. It has been found that after five cycles of caustic rinsing, the M4 module performance increased to 86% of its maximum value. However, the subsequent membrane rinsing and water filtration for 30 min did not allow improving the cleaning efficiency. This result is different from that obtained for PP membranes, where, after rinsing with NaOH solution, additional water filtration ensured the recovery of initial membranes permeability (Figure 9). In turn, the efficiency of the M4 module slightly increased after backwashing (25 min total, Figure 12, operations 10–14) and only membrane rinsing with 3% H_3_PO_4_ solution for 50 min allowed recovery of the initial module performance (Figure 12, operation 15–20). 

The cleaning time determined for individual rinsing solutions (Figure 12) was applied for the procedure of the M4 module cleaning (Figure 13) after the filtration was performed at the feed flow velocity equal to 11 m/s (Figure 7). The efficiencies obtained for each stage of membrane cleaning, presented in Figure 13, were similar to the previous ones (Figure 12). However, the rinsing of the membrane with 3% NaOH solution for 25 min provided a lower recovery of the membrane performance, equal to 69% of its maximum value (instead of 86%). In turn, after 25 min of backwash, 71% of the maximum permeate flux has been noted. It has been determined that final cleaning with H_3_PO_4_ solution for 50 min ensured the complete recovery of the maximum module performance (Figure 13, operation 4).

The use of the developed membrane cleaning procedure allowed to maintain the steady-state relative permeate flux equal to 0.2 for repeated MF process (Figure 14). A slight decrease in the initial permeate flux was observed, which was reduced by increasing the cleaning temperature from 30 °C to 45 °C and extending the contact time with the cleaning solutions (NaOH and H_3_PO_4_) to 60 min (Figure 14, series S6 and S7).

Summarizing, results obtained in the present study demonstrated that ceramic membranes can be effectively cleaned by suitable cleaning agents with selected concentration and duration of action. In turn, the use of aggressive solutions could cause rapid degradation of the polypropylene membrane matrix.

### 3.3. Filtration of Fermentation Broths with Lactobacillus casei Bacteria

It is well known that the intensity of membrane fouling depends on the composition and properties of a feed solution. Therefore, for comparative purposes, in the next step of our research, the MF of 1,3-PD fermentation broth with *Lactobacillus casei* was investigated. This feed contained more protein components than the broth with *Citrobacter freundii* (Table 1). It has been found that during the MF process of broth with *Lactobacillus casei* the reported relative flux was equal to 0.2 (Figure 15, S1 and S2). Thus, it was 10 percentage points lower than that obtained during the MF of broth with *Cirobacter freundii* (Figure 7), performed under the same values of TMP and V_F_, equal to 0.08 MPa and 5.5 m/s, respectively. These results indicate that increasing the concentration of foulants, such as meat and yeast extracts as well as peptone K in a feed, led to more intensive membrane fouling.

As expected, cleaning the M4 module with distilled water did not significantly affect the performance of the process and the relative permeate flux remained at the value equal to 0.2 (Figure 15, S1 and S2). 

A much higher module performance, with the stabilization of the relative permeate flux of 0.25, was obtained as a result of cyclic cleaning of the module with NaOH and acid solutions (Figure 15, Series 3–5). However, despite an extensive cleaning procedure, the performance of the M4 module only increased to 0.83 of its maximum value (Figure 16).

A similarly significant decline in the MF process performance with using PP membranes has been noted (Figure 17). Although the new M2 module for this investigation was used, the relative permeate flux was stabilized at 0.05 (Figure 17, S1). Membrane cleaning with distilled water led to a slight increase of the flux, however, the continued process of broth clarification led to the decline in relative flux again to the value of 0.05 (Figure 17, S2). For the next cleaning, 1% NaOH solution for 30 min was used, which allowed the recovery of the initial membrane performance. As a consequence, a lower decrease in the permeate flux during the next series of the MF process was obtained (Figure 17, S3). 

A series of MF experiments conducted allowed for comparison the performances of polypropylene membranes used for the clarification of fermentation broths with *Citrobacter freundii* (module M1, Figure 9) and *Lactobacillus casei* (module M2, Figure 17). It was found that although the more favorable operating conditions were used for the process with the M2 module, a decrease in the permeate flux was greater. This finding confirmed the above-mentioned result indicating that the broth composition plays a key role in the intensity of membrane fouling, and the increased concentration of protein components in the feed leads to a greater reduction of the filtration process performance.

### 3.4. Filtration of Yeast Suspensions

It is well known that the design of the module and the turbulence of the feed low affect the intensity of the membrane fouling phenomenon. Hence, in the present study, the M3 and M4 modules with the same design and similar membranes diameters (Table 2) were applied for clarification of *Saccharomyces cerevisiae* yeast suspensions in the pilot installation (Figure 1b). It ensured investigating the M3 and M4 modules performance under similar hydrodynamic conditions. 

Although the high flow velocities (Re number = 12,000) for the process with using the M3 module have been applied, during the MF process of the solution containing 0.1% yeast, the relative permeate flux decreased to a value below 0.06 (Figure 18, S1). It was noticed that rinsing the module with distilled water caused a slight increase in the membrane performance, while the use of chemical cleaning agents (1% NaOH for 20 min and 0.5% H_3_PO_4_ for 20 min) led to an increase of the relative flux to 0.8. However, continuation of the MF process with the membrane cleaned by this method resulted in a greater decline in the performance, below 0.04 of the maximum permeate flux (Figure 18, S2). It is worth noting that, typically, yeast cells are 5–10 µm in diameter [87,88], hence, they are retained by the membranes used in our research. However, the prepared feed solution, apart from yeast cells, contained other substances, e.g., proteins, which formed a deposit layer on the surface of the membranes (Figure 19). The presence of additional components resulted from the fact that freeze-dried distiller’s yeast was used in the study.

In order to reduce the intensity of membrane fouling, backwash (1 min) every 10 min was performed. As a result, a two-fold increase in MF process performance was obtained (Figure 18, S3–S5). The discontinuation of backwash resulted in a significant drop in the module performance (Figure 18, S6). Increasing the yeast content in the feed led to a decrease of the process performance, although for both 0.16% and 0.5% solutions the declines in permeate flux were similar (Figure 18, S7 and S8). This finding is in agreement with that presented by Hassan et al. [20] who indicated that during the MF of *Saccharomyces cerevisiae* yeast suspensions, the steady-state permeate flux is independent from the cell initial concentration in the range from 6 to 10 g/L. 

To limit decrease in the flux, backwashing again was applied which resulted in a slight increase in the performance (Figure 18, S8–S10). Indeed, the observed increases were not as significant as for the series S3–S5, also for the solution containing 0.1% yeast (Figure 18, S10). It should be emphasized that after each chemical cleaning, the module performance was recovered, which was even greater than the initial performance (Figure 20, CR1–CR5). SEM studies showed that such an increase could be due to slight degradation of the membrane matrix. Likewise to the previous finding (Figure 11), it was determined that the membrane surface porosity increased significantly. Indeed, the surface pores of 2–5 µm were observed (Figure 21a). In this case, after starting the MF process, the pores quickly filled with filtered biomass, which also resulted in a rapid decline in the permeate flux. Indeed, the penetration of biomass into the pores was confirmed by the observations of the membrane cross-section (Figure 21b). Despite the use of cleaning the membrane with alkaline and acid solutions, deposits were still found inside the pores up to a depth of 2 µm.

Regarding the ceramic membrane used, its performance was also very low. The noted relative permeate flux, similar to the PP membrane, was equal to 0.045 (Figure 22, S2). Although ceramic membranes exhibit good chemical resistance, it must be recognized that the use of the same chemical agents as for PP membrane (NaOH and H_3_PO_4_) was not efficient and led to a decline in the module performance after the first cleaning cycle (Figure 23, 1–3). Hence, additional operations were required. Finally, the initial membrane performance by cleaning with the HNO_3_ solution as a final step was recovered. 

Cleaning of the ceramic membrane carried out after the second series of MF measurements (Figure 22 S2) showed that HNO_3_ solution is not sufficient to effectively clean the membrane, since its double use allowed to obtain the relative permeate flux equal to 0.7 (Figure 24). On the other hand, the use of NaOH solution with hypochlorite allowed increasing the value to 0.8, while the complete recovery of the membrane permeability was obtained after repeated cleaning with NaOH and H_3_PO_4_ solutions.

To sum up, our investigations have shown that fermentation broth components from ethanol production led to the intense membranes fouling, the removal of which required multi-step cleaning. It has been indicated that ceramic membranes, contrary to PP membranes, can be cleaned with aggressive solutions without any damage to the membrane matrix. 

## 4. Conclusions

This paper outlines a direct comparison of efficiency, performance and susceptibility to cleaning of the ceramic and polymeric MF membranes applied for clarification of 1,3-propanediol fermentation broths and suspensions of yeast *Saccharomyces cerevisiae*. For this purpose, ceramic membrane made of TiO_2_ (hydrophilic) and polypropylene (hydrophobic) membranes, in different modules configurations (tubular and capillary), were tested. The MF experiments were carried out at a constant temperature of 30 °C and a wide range of process parameters. The transmembrane pressure ranging from 0.02 to 0.08 MPa was used. In turn, the applied feed flow velocity was from 0.8 to 11 m/s. 

Both the ceramic and PP membranes have demonstrated a very high efficiency in purifying the biological suspensions. Although the number of bacteria in the feed was of an order of 12 log CFU, the membranes ensured sterile permeate during the filtration of 1,3-PD post-fermentation solutions. Although during the broths separation the turbidity of the feed increased to 8000 NTU, the turbidity of the obtained permeate was in the range of 1–3 NTU. 

Moreover, the results obtained in the present study offer vital evidence that the permeate quality is directly correlated to the membrane pore size. Indeed, since the ceramic membrane had a nominal pore size (0.14 µm) smaller than the PP membranes (0.20 µm), it allowed obtaining the permeate characterized by a lower turbidity, equal to 0.2 NTU. 

It has been shown that all the membranes used were very susceptible to fouling, both by the components of 1,3-PD fermentation broths and yeast suspensions. The significant impact of the feed flow velocity and the composition of the fermentation broths on the relative permeate flux has been pointed out. It has been demonstrated that the membrane properties (hydrophobic/hydrophilic) did not significantly affect the intensity of the fouling phenomenon. The obtained data show that the decrease in process performance was mainly dependent to the properties of the filter cake formed on the membranes surface. For the ceramic membrane, internal fouling also had a significant influence on the permeate flux decline.

Moreover, much attention has been paid to the development of effective and comprehensive cleaning procedures for the membranes used. It has been demonstrated that suitable cleaning agents with selected concentration and duration of action effectively cleaned both the ceramic and PP membranes. However, the use of aggressive solutions led to degradation of the PP membranes matrix. 

Findings of this study add to a growing body of literature on the use of ceramic and polypropylene MF membranes for the clarification of biological materials. However, it should always be taken into account that the fermentation broths are very complex medium, the composition of which strictly depends on the carbon source, bacteria strain and fermentation conditions as well as several other factors. Therefore, the results of the presented work give an overall view of the MF process efficiency for biological suspensions, however, all the characteristic dependencies should be examined individually.

## Figures and Tables

**Figure 1 membranes-11-00044-f001:**
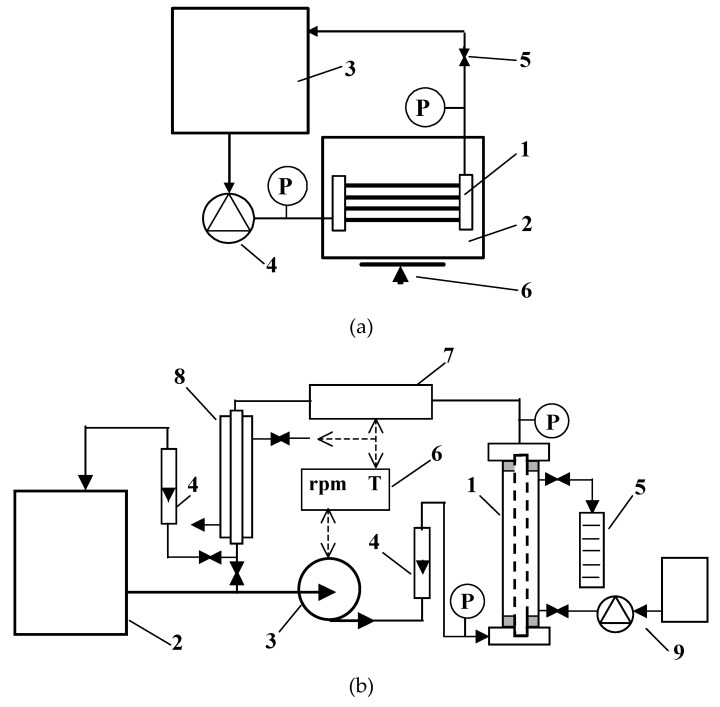
Experimental installations: (**a**) Laboratory trial: 1—polypropylene (PP) module, 2—permeate tank, 3—feed tank, 4—peristaltic pump, 5—valve, 6—balance, P—manometer; (**b**) pilot plant: 1—tubular module, 2—feed tank, 3—pump, 4—rotameter, 5—measurement cylinder, 6—temperature and rpm (pump) regulator, 7—electrical heater, 8—water cooling, 9—backflushing system, P—manometer.

**Figure 2 membranes-11-00044-f002:**
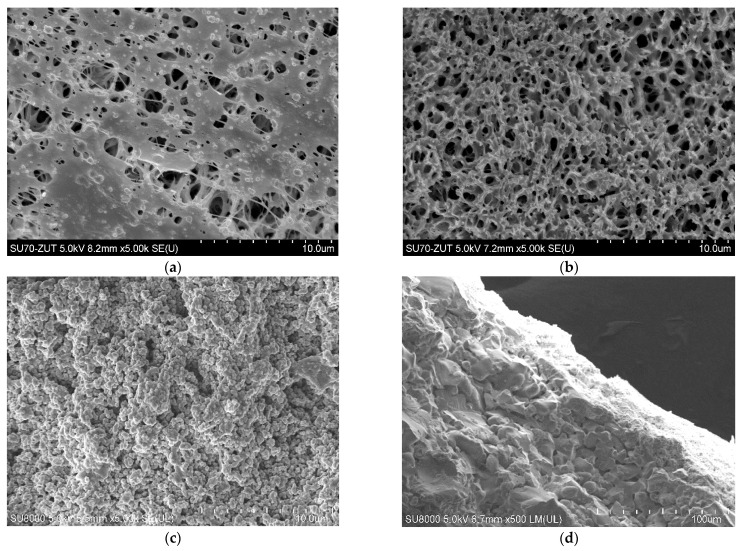
SEM images of tested membranes. Accurel PP S6/2 membrane: (**a**) surface, (**b**) cross-section, ceramic membrane: (**c**) surface and (**d**) cross-section.

**Figure 3 membranes-11-00044-f003:**
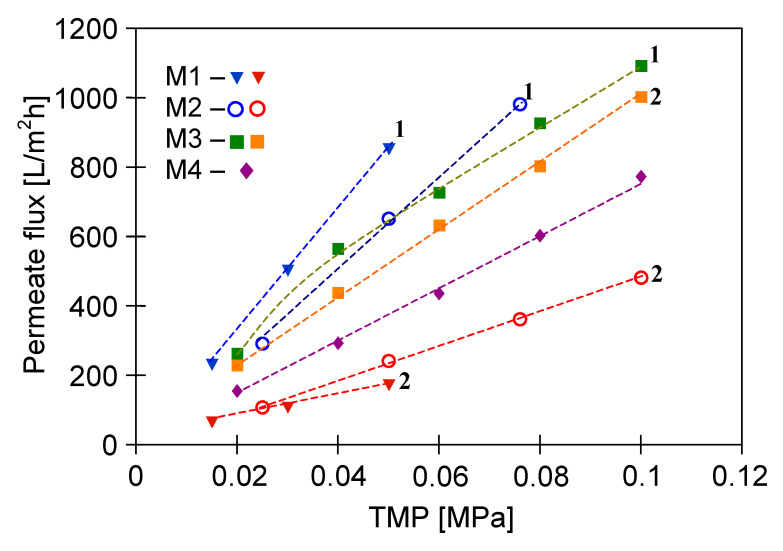
The maximum permeate flux as a function of transmembrane pressure (TMP): 1—after one operation of membrane wetting, 2—stabilized J_0_ values after several operations of membrane wetting.

**Figure 4 membranes-11-00044-f004:**
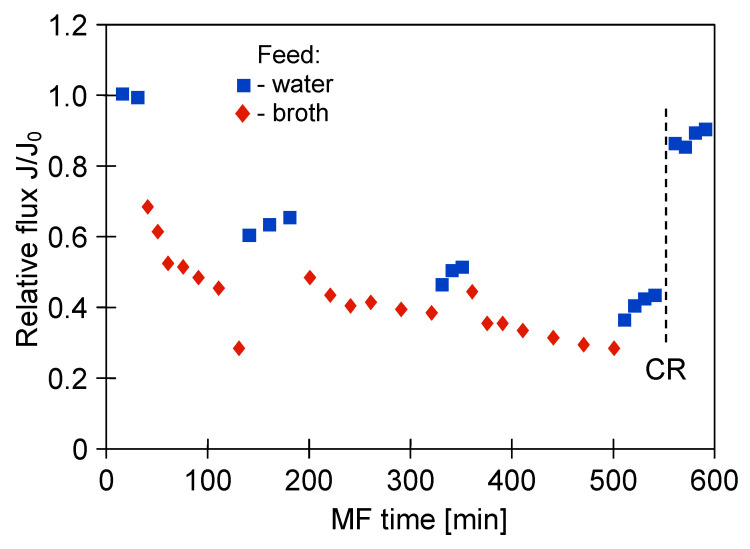
Changes in relative permeate flux during microfiltration (MF) of 1,3-propanediol (1,3-PD) fermentation broth with *Citrobacter freundii*. Module M1, TMP = 0.03 MPa, V_F_ = 0.8 m/s, CR—chemical cleaning (1% NaOH).

**Figure 5 membranes-11-00044-f005:**
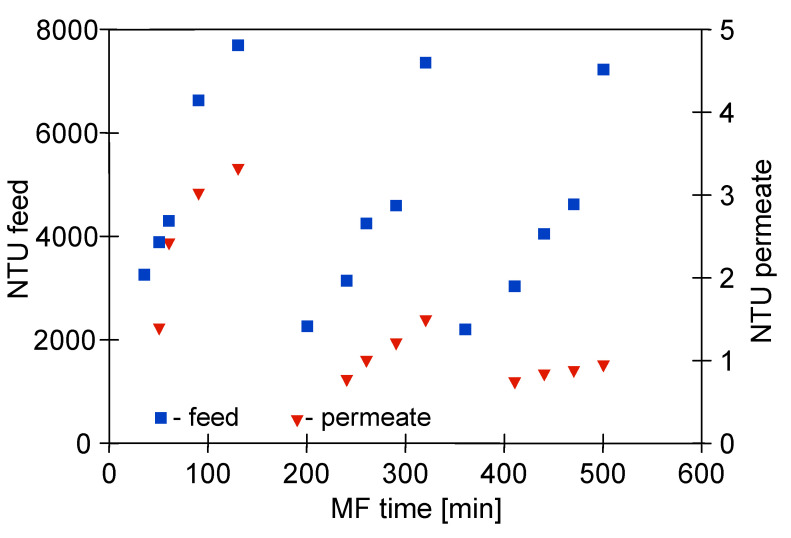
Changes in feed and permeate turbidity during MF of 1,3-PD fermentation broth with *Citrobacter freundii*. Module M1, TMP = 0.03 MPa, V_F_ = 0.8 m/s.

**Figure 6 membranes-11-00044-f006:**
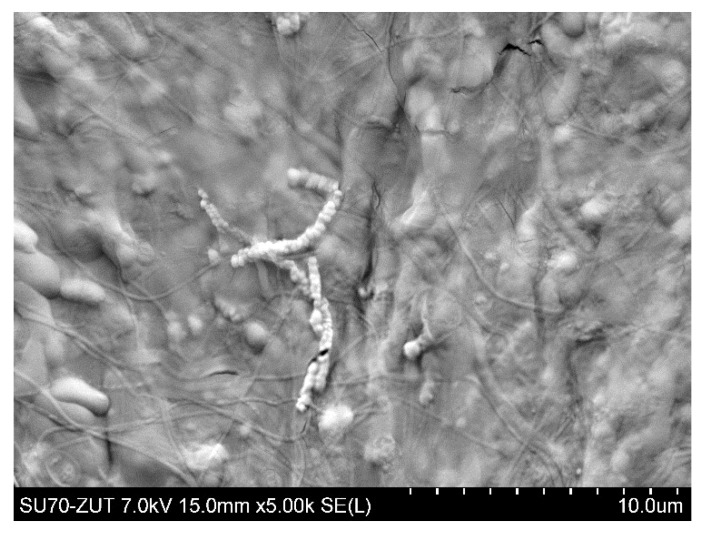
SEM image of PP S6/2 membrane (module M1) surface covered by deposit after MF process of fermentation broth with *Citrobacter freundii*. Deposit formed during 6 h of MF process.

**Figure 7 membranes-11-00044-f007:**
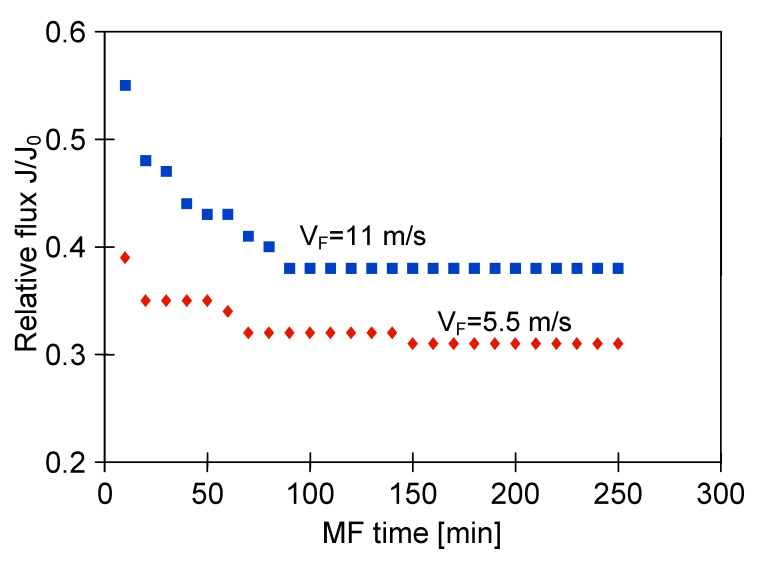
Effect of feed flow velocity on the relative permeate flux during MF of 1,3-PD fermentation broth with *Citrobacter freundii*. Module M4, TMP = 0.08 MPa, J_0_ = 311 l/m^2^h.

**Figure 8 membranes-11-00044-f008:**
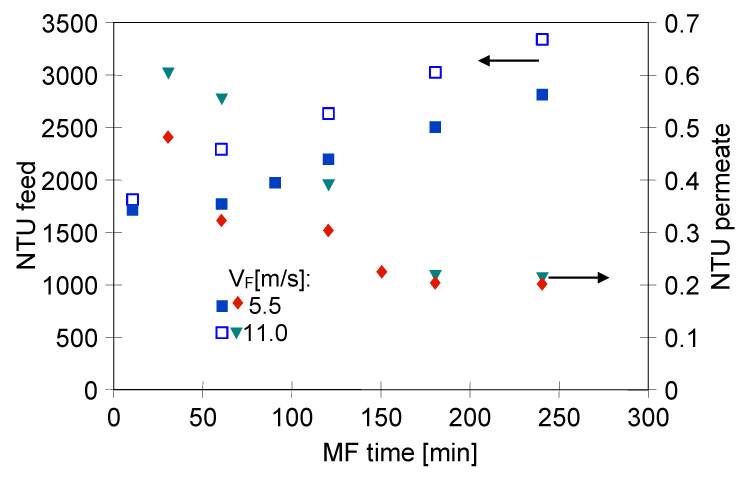
Changes in feed and permeate turbidity during MF of 1,3-PD fermentation broth with *Citrobacter freundii*. Module M4, TMP = 0.08 MPa.

**Figure 9 membranes-11-00044-f009:**
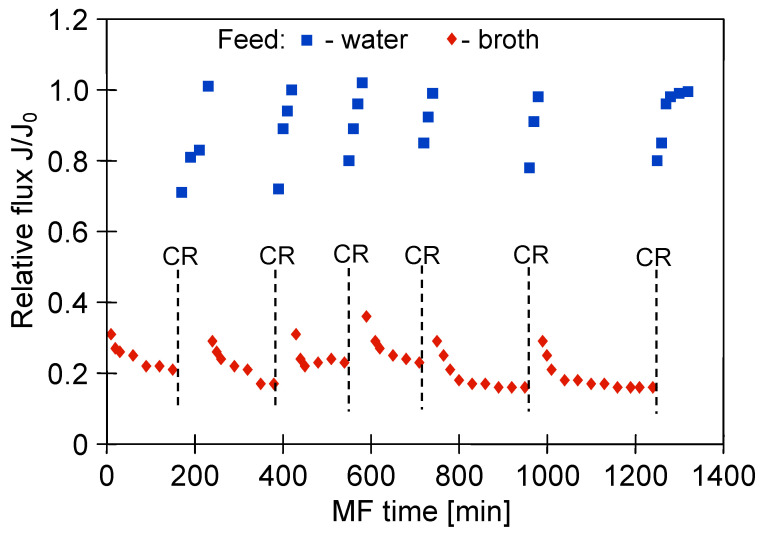
Changes in relative permeate flux during the long-term MF process of 1,3-PD fermentation broth with *Citrobacter freundii*, with periodically cleaning the module with 1% sodium hydroxide (NaOH) solution. Module M1, TMP = 0.03 MPa, V_F_ = 0.8 m/s, CR—chemical cleaning.

**Figure 10 membranes-11-00044-f010:**
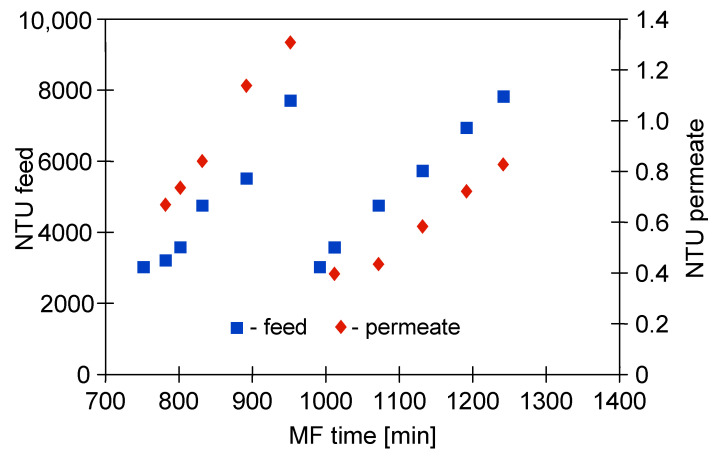
Changes in feed and permeate turbidity during the last two measurement series of the MF of 1,3-PD fermentation broth with *Citrobacter freundii*. Module M1, TMP = 0.03 MPa, V_F_ = 0.8 m/s.

**Figure 11 membranes-11-00044-f011:**
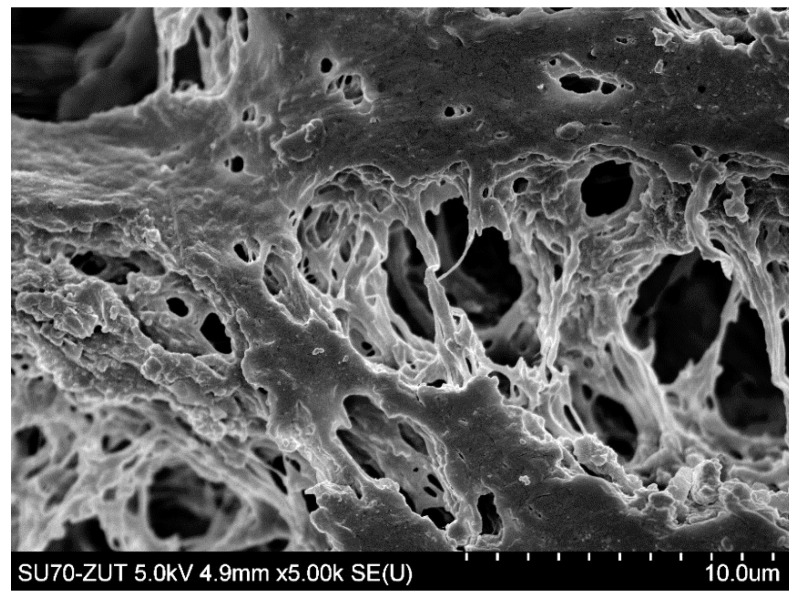
SEM image surface of Accurel PP S6/2 membrane. Module M1—membranes several times rinsed with 1% NaOH solution.

**Figure 12 membranes-11-00044-f012:**
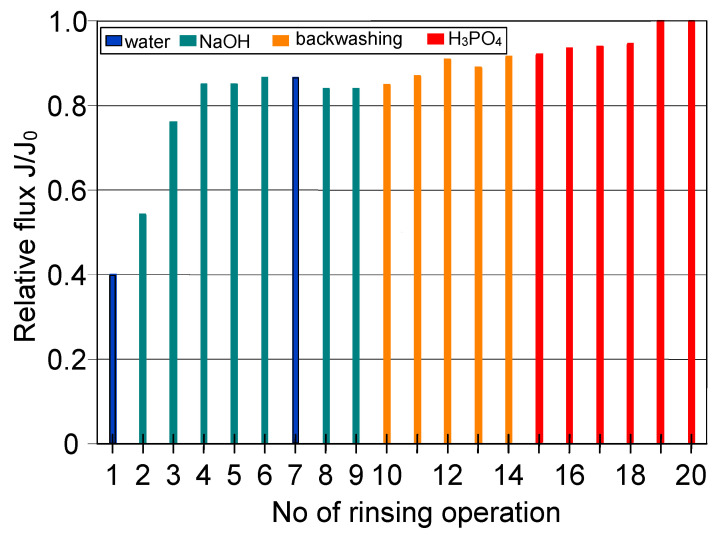
The efficiency of cleaning the ceramic membrane after the MF of 1,3-PD fermentation broth with *Citrobacter freundii* (V_F_ = 5.5 m/s. TMP = 0.08 MPa). Operation time: water—30 min, NaOH—5 min, backwash—5 min, acid—10 min.

**Figure 13 membranes-11-00044-f013:**
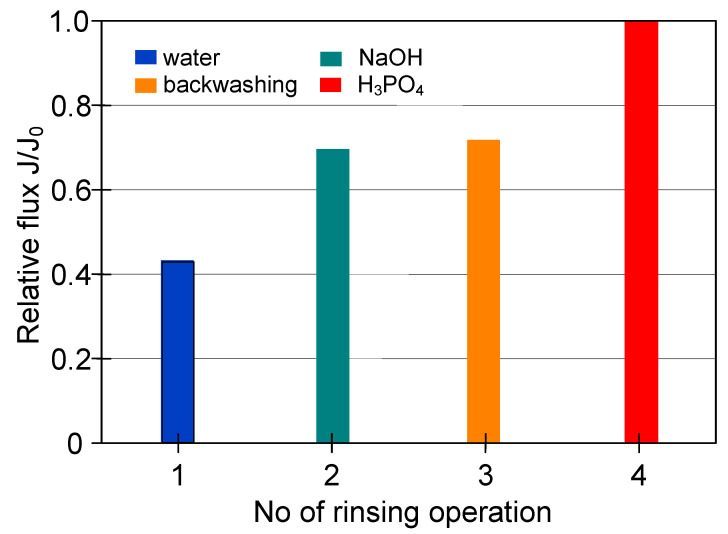
The efficiency of cleaning the ceramic membrane after the MF of 1,3-PD fermentation broth with *Citrobacter freundii* (V_F_ = 11 m/s. TMP = 0.08 MPa). Operation time: water—30 min, NaOH—25 min, backwash—25 min, acid—50 min.

**Figure 14 membranes-11-00044-f014:**
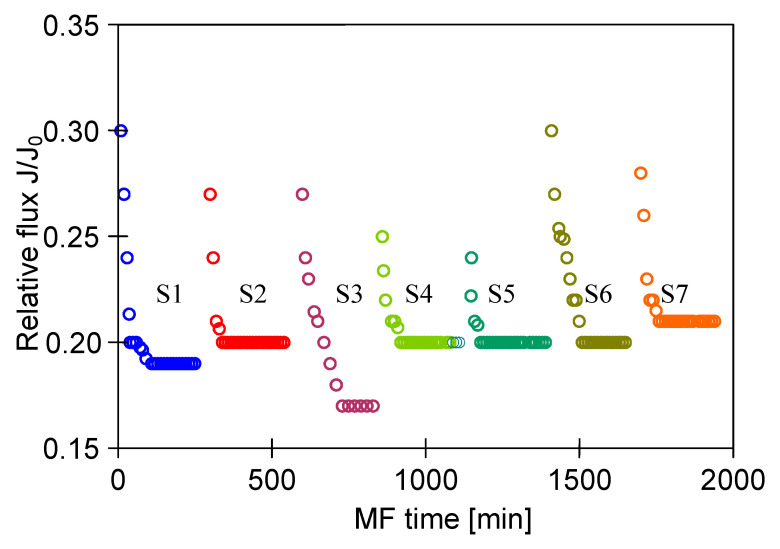
Changes in relative permeate flux during the long-term MF process of 1,3-PD fermentation broth with *Citrobacter freundii*. Module M4, TMP = 0.02 MPa, V_F_ = 5.5 m/s, CR—chemical cleaning.

**Figure 15 membranes-11-00044-f015:**
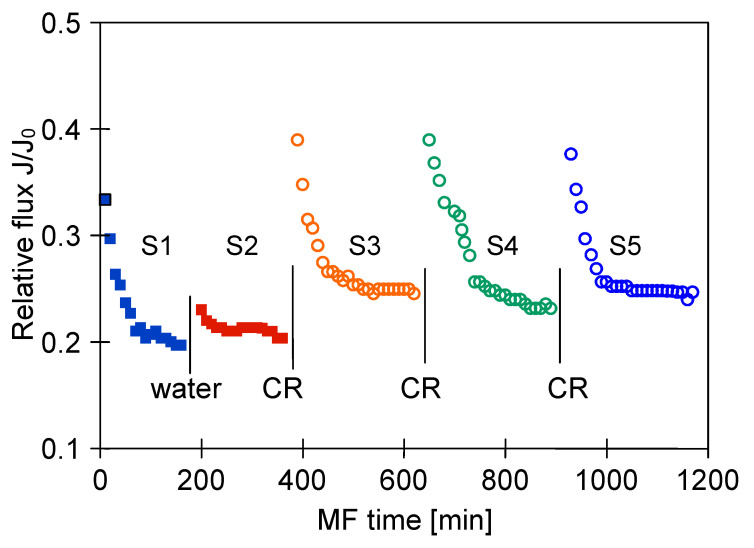
Changes in relative permeate flux during the long-term MF process of 1,3-PD fermentation broth with *Lactobacillus casei* with periodically cleaning the module. Module M4, TMP = 0.08 MPa, V_F_ = 5.5 m/s, CR—chemical cleaning.

**Figure 16 membranes-11-00044-f016:**
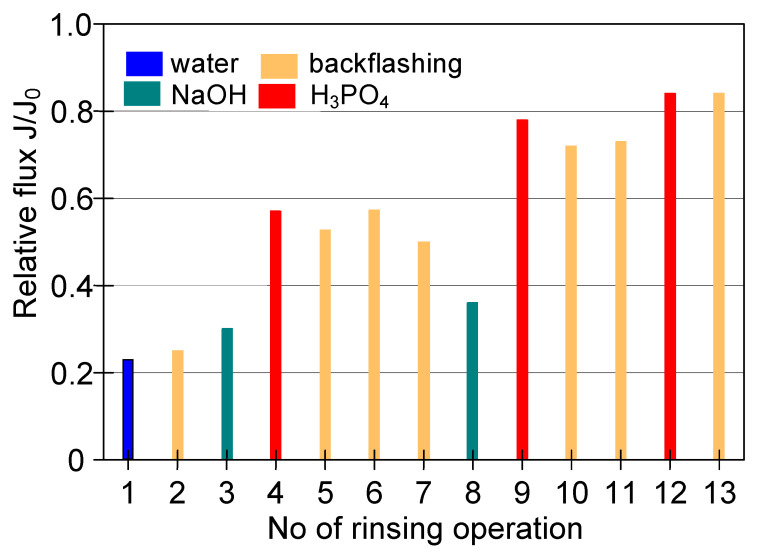
The efficiency of cleaning the ceramic membrane after the MF (TMP = 0.08 MPa, V_F_ = 5.5 m/s) of 1,3-PD fermentation broth with *Lactobacillus casei*. Operation time: water—30 min, backwash—5 min, NaOH—60 min, acid—60 min.

**Figure 17 membranes-11-00044-f017:**
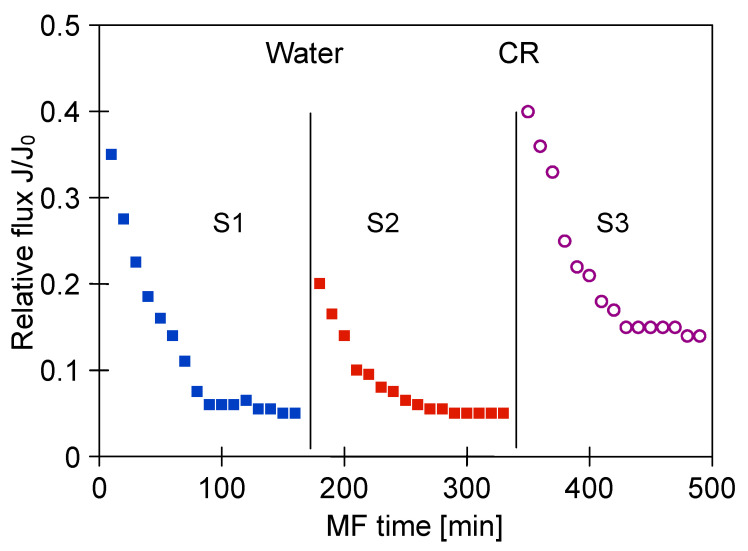
Changes in relative permeate flux during the long-term MF process of 1,3-PD fermentation broth with *Lactobacillus casei* with periodically cleaning the module. Module M2, TMP = 0.05 MPa, V_F_ = 1.5 m/s, CR—chemical cleaning (NaOH).

**Figure 18 membranes-11-00044-f018:**
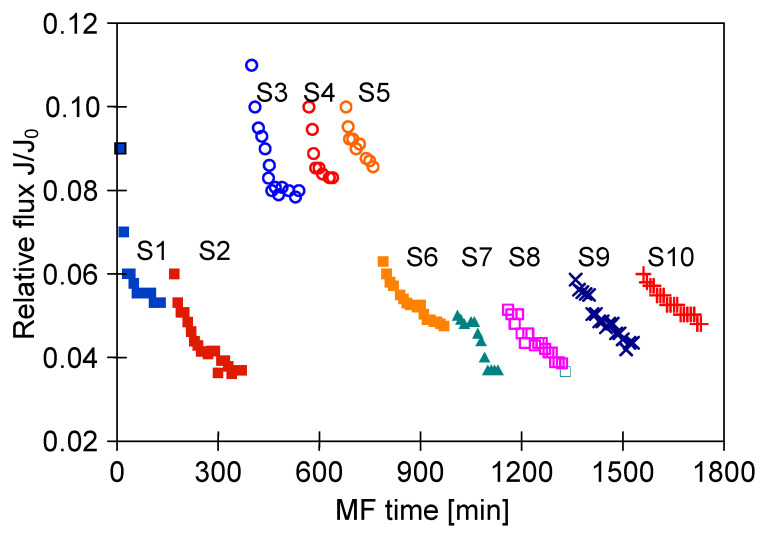
Changes in relative permeate flux during the long-term MF process of yeast suspension. Module M3, TMP = 0.08 MPa, V_F_ = 5.5 m/s.

**Figure 19 membranes-11-00044-f019:**
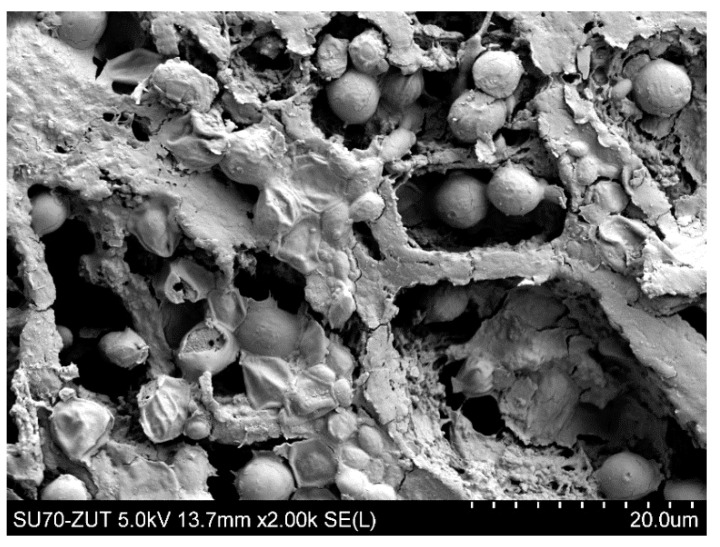
SEM image of PP S6/2 membrane (module M3) surface covered by biological deposit with yeast cells after MF process of yeast suspensions.

**Figure 20 membranes-11-00044-f020:**
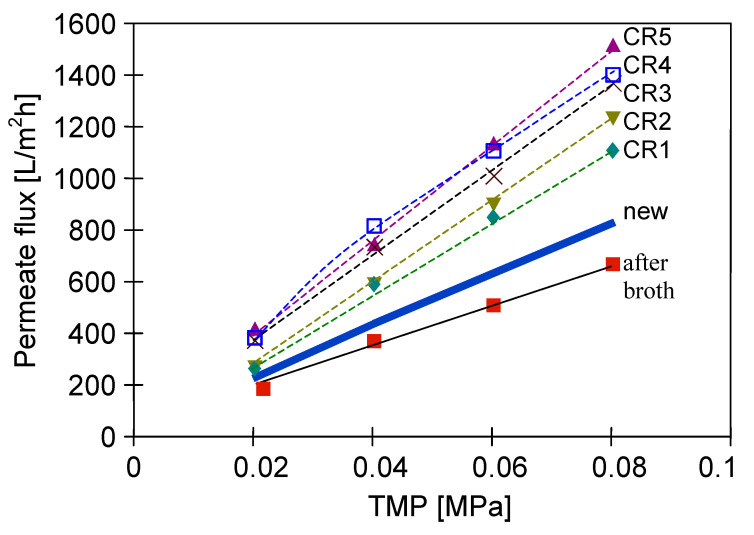
Effect of chemical membrane cleaning on the maximum permeate flux. Module M3. CR—chemical rinsing.

**Figure 21 membranes-11-00044-f021:**
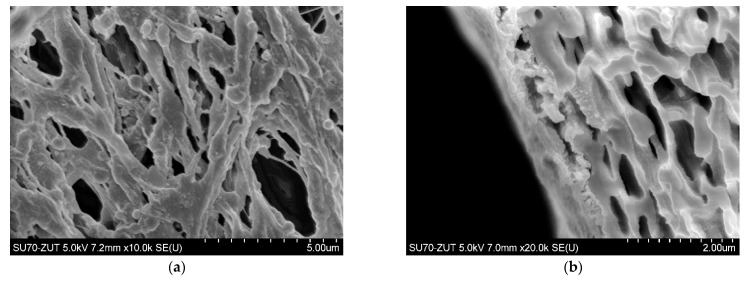
SEM images of PP membranes after cyclic chemical rinsing. (**a**) membrane surface, (**b**) membrane cross-section with deposit inside surface pores.

**Figure 22 membranes-11-00044-f022:**
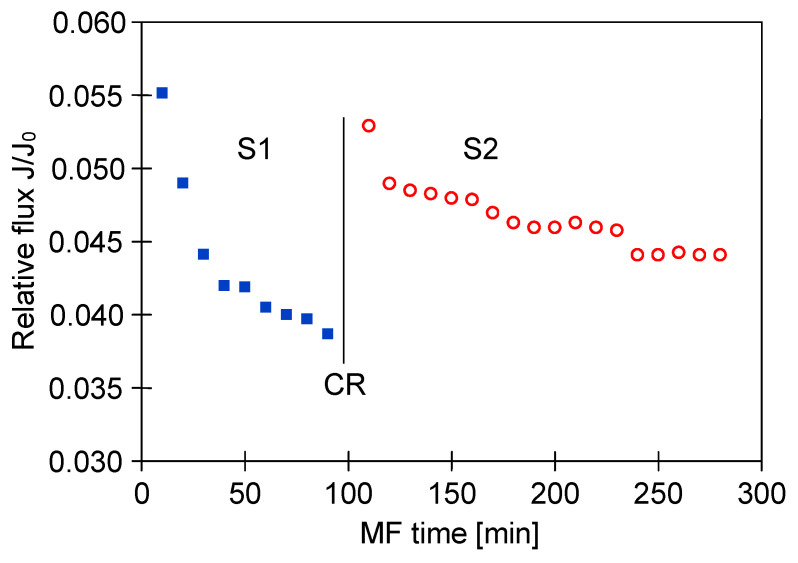
Changes in relative permeate flux during the MF process of yeast suspension with periodically cleaning the module. Module M4, TMP = 0.08 MPa, V_F_ = 5.5 m/s, CR—chemical cleaning.

**Figure 23 membranes-11-00044-f023:**
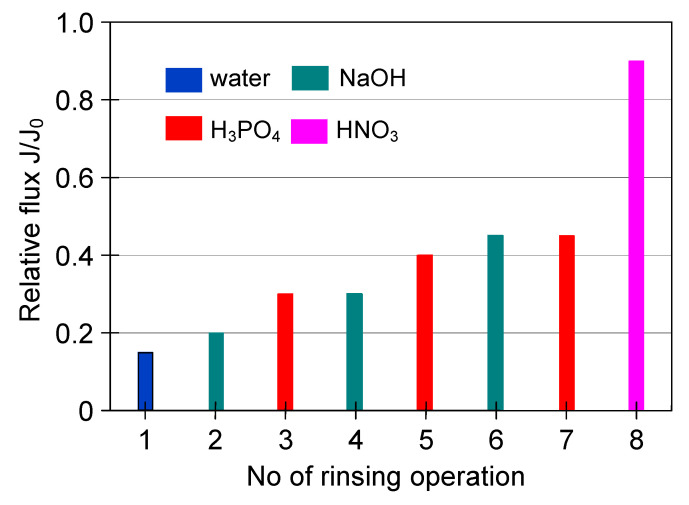
The efficiency of cleaning the ceramic membrane after the MF of yeast suspension (the first series of measurements). Operation time: water—30 min, 2% NaOH—40 min, 1% H_3_PO_4_—40 min, 2% HNO_3_—30 min.

**Figure 24 membranes-11-00044-f024:**
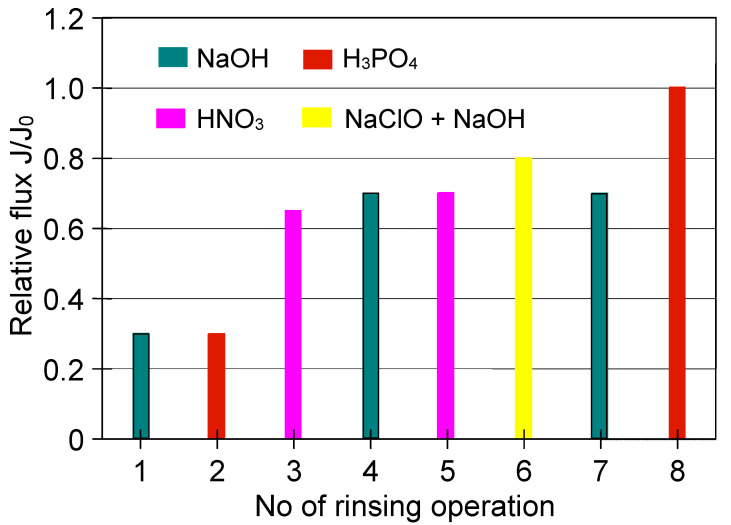
The efficiency of cleaning the ceramic membrane after the MF of yeast suspension (the second series of measurements). Operation time: 2% NaOH—60 min, 1% H_3_PO_4_—30 min, 2% HNO_3_—30 min, 2 L of 1% NaOH + 200 mL NaClO—60 min.

**Table 1 membranes-11-00044-t001:** The compositions of post-fermentation broths with *Lactobacillus casei* and *Citrobacter freundii* bacteria.

Component [g/L]	*Lactobacillus casei*	*Citrobacter freundii*
wetted mass	21.6–27.9	28–32.5
meat extract	8	1.5
yeast extract	5	2
peptone K	10	2.5
lactic acid	13.5–15.2	2.1–3.5
1,3-propanediol	-	18.9–22.5
glycerol	3–4.7	1.7–3.8
acetic acid	2–3.6	1.2–2.3
formic acid	0.25–0.3	0.8–1.22
succinic acid	-	0.7–2.13
amonium citrate	2	-
K_2_HPO_4_·3H_2_O	0.6	3.4
KH_2_PO_4_	-	1.3
MgSO_4_·7H_2_O	0.4	0.9
(NH_4_)_2_SO_4_	-	2
CH_3_COONa	1.5	-
CaCl_2_·2H_2_O	-	0.01
CoCl_2_·6H_2_O	0.004	0.002

**Table 2 membranes-11-00044-t002:** Characteristics of the MF membranes used in the experiments—manufacturers data.

Module	Manufacture	Membrane	d^1^ [µm]	D^2^ [mm]	Length [cm]	Wall [mm]	Area [cm^2^]
M1, M2	Membrana GmbH, Germany	Accurel PP S6/2	0.20	1.8	95	0.4	21.5
M3	Membrana GmbH, Germany	Accurel PP V8/2 HF	0.20	5.5	22	1.5	38.0
M4	Tami Ind., France	TiO_2_	0.14	5.6	22	2.0	38.7

^1^ Pore diameter. ^2^ Capillary membrane inner diameter.

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
