# Peer review of "Comparison of Polypropylene and Ceramic Microfiltration Membranes Applied for Separation of 1,3-PD Fermentation Broths and Saccharomyces cerevisiae Yeast Suspensions"

_membranes, 2021, doi:10.3390/membranes11010044_

Round 1

Reviewer 1 Report

Comments

The subject matter is suitable for publication in Membranes.

I have read the article very carefully and it is, in general, an interesting paper because the authors made a comparison of polypropylene and ceramic microfiltration membranes applied for the separation of 1,3-propanediol fermentation broths and yeast suspensions.

The objective of this study is the direct comparison of efficiency, performance and susceptibility to cleaning of the ceramic and polymeric microfiltration membranes applied for the clarification of 1,3-propanediol fermentation broths and aqueous suspensions of dry distillery yeast Saccharomyces cerevisiae-Bc 16a. The authors study two different modules configuration (tubular and capillary).

The manuscript contains a lot of information and should be reorganized.

Although the authors carry out a very broad study of the fouling and cleaning of the different microfiltration membrane, along the manuscript the information is not well presented. Therefore, the authors should, in the conclusion part, extract more information from the study carried out. The conclusion section should contain a response for the objectives determined in the manuscript.

The different sections of the manuscript are neither clear nor well-structured. The authors in some sections should compare their results with those of other experimental investigations and discuss them comparatively.

To improve the understanding of the manuscript, abbreviations section should be added.

Anyway, I include some specific comments:

  1. Materials and Methods

Table 1, please use the subscripts in the nomenclature of chemical formulas.

Table 2, please change “membrana” by “membrane”

2.3. Analytical methods

In page 4 line 213 the authors say ..... “Permeate and feed samples were analyzed in terms of compounds content, turbidity and number of bacteria.”

Why do the authors use there parameters?

In the following figures (figures 5, 8 and 10) turbidity is the only parameter shown, not the others.

The authors should justify the selection of this procedure.

Were the experimental assays made in duplicate? What is the reproducibility of the experimental results?

  1. Results

In order to organize this section, Figures 4 and 5 should be in one Figure (M1), similar to Figures 7 and 8 (M4), Figures 9 and 10, and figures 12 and 13.

Although the objective of this study is to compare the different microfiltration membranes and module configuration, the results are described according to the type of fermentation used. The authors should simplify these sections and give more relevance to the behavior of the membranes after cleaning.

References

The number of references is too high (112) to be a research article. Authors should select only citations that are relevant in the text.

I do not think the paper is acceptable in its present form. It needs a major revision.

Author Response

Dear Editor and Reviewers,

We thank Reviewers for their interest in our work and the valuable comments.

Since two Reviewers indicated that ‘English language and style are fine/minor spell check required’ (the third Reviewer indicated that does not feel qualified to judge about the English language and style), we have made some minor changes to the English language. Moreover, we have improved the manuscript accordingly to the Reviewer’s indications. All changes made in the manuscript are marked in blue.

Please find below the responses to all comments provided by three Reviewers.

Thank you for your time and effort.

Sincerely,

The authors

Review 1

The subject matter is suitable for publication in Membranes. I have read the article very carefully and it is, in general, an interesting paper because the authors made a comparison of polypropylene and ceramic microfiltration membranes applied for the separation of 1,3-propanediol fermentation broths and yeast suspensions. The objective of this study is the direct comparison of efficiency, performance and susceptibility to cleaning of the ceramic and polymeric microfiltration membranes applied for the clarification of 1,3-propanediol fermentation broths and aqueous suspensions of dry distillery yeast Saccharomyces cerevisiae-Bc 16a. The authors study two different modules configuration (tubular and capillary).

1)    The manuscript contains a lot of information and should be reorganized.

While we appreciate the Reviewer’s feedback, we respectfully disagree with this comment. Indeed, the manuscript contains a lot of information, however, it is well organized and the results are presented in logical order, as follows:

3.1. Membranes morphology and maximum performance

3.2. Filtration of fermentation broths with Citrobacer freundii bacteria

3.3. Filtration of fermentation broths with Lactobacillus casei bacteria

3.4. Filtration of yeast suspensions

The first section deals with the properties of the membranes used, while each of the following sections relates to different separated solutions. We believe that presentation of results ordered depending on the type of the separated solution ensures the best readability. Moreover, we would like to emphasize that none of the other two Reviewers indicated that the article is not well organized.

Moreover, in order to underline the selected order of the presented results, we have added an explanation at the end of chapter 3.1.

-   Lines 261-266: Fig. 3 shows the stabilized performances obtained for the tested modules fed with distilled water. Unfortunately, the permeate flux decreases significantly during filtration of real solutions. Importantly, the intensity of the flux decline, apart from the properties of membranes (hydrophilic or hydrophobic), is significantly influenced by the composition of the feed. For this reason, in order to investigate the intensity of the fouling phenomenon, in subsequent stages of the research, in addition to yeast suspension, real broths with different compositions (Table 1) were used.

2)    Although the authors carry out a very broad study of the fouling and cleaning of the different microfiltration membrane, along the manuscript the information is not well presented. Therefore, the authors should, in the conclusion part, extract more information from the study carried out. The conclusion section should contain a response for the objectives determined in the manuscript.

Thank you for the suggestion. We agree with the Reviewer’s comment, that the conclusion section should contain a response for the objectives determined in the manuscript. The objective of this study is the direct comparison of efficiency, performance and susceptibility to cleaning of the ceramic and polymeric microfiltration membranes applied for the clarification of 1,3-propanediol fermentation broths and aqueous suspensions of dry distillery yeast Saccharomyces cerevisiae-Bc 16a.

Therefore, we have improved the conclusion section by demonstrating the following comparisons:

Similarities:

-      Both the ceramic and PP membranes have demonstrated a very high efficiency in purifying the biological suspensions. Although the number of bacteria in the feed was of an order of 12 log CFU, the membranes ensured sterile permeate during the filtration of 1,3-PD post-fermentation solutions. Although during the broths separation the turbidity of the feed increased to 8000 NTU, the turbidity of the obtained permeate was in the range of 1-3 NTU. (lines 586-590),

-      It has been shown that all the membranes used were very susceptible to fouling, both by the components of 1,3-PD fermentation broths and yeast suspensions. The significant impact of the feed flow velocity and the composition of the fermentation broths on the relative permeate flux has been pointed out. It has been demonstrated that the membrane properties (hydrophobic/hydrophilic) did not significantly affect the intensity of the fouling phenomenon. The obtained data show that the decrease in process performance was mainly dependent to the properties of the filter cake formed on the membranes surface. For the ceramic membrane, internal fouling also had a significant influence on the permeate flux decline. (lines 595-602).

Differences:

-      Indeed, since the ceramic membrane had a nominal pore size (0.14 µm) smaller than the PP membranes (0.20 µm), it allowed obtaining the permeate characterized by a lower turbidity, equal to 0.2 NTU (lines 592-594),

-      It has been demonstrated that suitable cleaning agents with selected concentration and duration of action effectively cleaned both the ceramic and PP membranes. However, the use of aggressive solutions led to degradation of the PP membranes matrix. (lines 604-607).

3)    The different sections of the manuscript are neither clear nor well-structured. The authors in some sections should compare their results with those of other experimental investigations and discuss them comparatively.

After correcting the article, the results of the present study have been compared with those of other experimental investigations, as follows:

-      lines 310-312,

-      lines 325-330,

-      lines 306-320,

-      lines 356-364.

-      lines 526-528.

4)    To improve the understanding of the manuscript, abbreviations section should be added.

We agree with the Reviewer’s comment, thus in order to improve the understanding of the manuscript, we have added ‘list of abbreviations’.

5)    Table 1, please use the subscripts in the nomenclature of chemical formulas.

In the submitted version of the article we have used the subscripts in the nomenclature of chemical formulas. However, the article file downloaded from the Membranes website is without the subscripts, we do not know why. Therefore, we have corrected it. Moreover, before printing, we will check the article proof very carefully and we will successfully eliminate all such errors.

6)    Table 2, please change “membrana” by “membrane”

The word ‘membrana’ is corrected since it is the company name ‘Membrana GmbH’.

7)     In page 4 line 213 the authors say ..... “Permeate and feed samples were analyzed in terms of compounds content, turbidity and number of bacteria.” Why do the authors use there parameters?

Primarily, turbidity and number of bacteria are parameters which determine the quality of the permeate obtained. Moreover, separation of fermentation products is a multi-stage process, whereas the microfiltration is proposed as a pre-treatment stage. In order to use NF / RO for further steps, it is necessary to remove turbidity and ensure sterility, otherwise there will be intense biofouling destroying membrane modules. In turn, the compounds content has been analysed, since it affects the membrane fouling.

We have improved the manuscript, as follows:

-      lines 138-141: In subsequent stages, metabolites are separated by the nanofiltration (NF) and reverse osmosis (RO) [73]. The spiral wound structure of membrane modules used in these processes requires careful removal of turbidity (Silt Density Index <5) and sterility of the obtained MF permeate.

-      lines L295-298: Considering the application of NF/RO processes for MF permeate separation, the sterility and turbidity of filtered fermentation broths are that controlling the efficiency of the MF process. Therefore, in the present study these quality attributes of the obtained permeate were continuously controlled.

8)    In the following figures (figures 5, 8 and 10) turbidity is the only parameter shown, not the others. The authors should justify the selection of this procedure.

Justification for selecting this procedure can be found in the text:

-      lines 138-141: In subsequent stages, metabolites are separated by the nanofiltration (NF) and reverse osmosis (RO) [73]. The spiral wound structure of membrane modules used in these processes requires careful removal of turbidity (Silt Density Index <5) and sterility of the obtained MF permeate.

-      lines 295-300: Considering the application of NF/RO processes for MF permeate separation, the sterility and turbidity of filtered fermentation broths are that controlling the efficiency of the MF process. Therefore, in the present study these quality attributes of the obtained permeate were continuously controlled. It has been found that both the ceramic and PP membranes used provided a sterile permeate. Indeed, although the number of bacteria in the feed was of an order of 12 log CFU, no bacteria was detected in the permeate samples.

-      lines 347-350: It has been found that, likewise to the process with using the PP membranes, the obtained permeate was sterile and its turbidity decreased over the MF time. It has been demonstrated that the ceramic membrane used provided obtaining the permeate characterized by the turbidity equal to 0.2 NTU at the end of the trial.

9)    Were the experimental assays made in duplicate? What is the reproducibility of the experimental results?

In the present study, 4 randomly selected capillaries with a length of about 1 m were installed in the M1 and M2 module, which is similar to the size of the industrial MF module. We conducted the experiments for several months, which allowed us to stabilize the parameters of the modules and demonstrate the actual operational capabilities of the process. As a result, we obtained a good repeatability of the measurements, which is assessed by the slopes of the initial performance drop and then changes in the steady-state permeate flux. For instance, in Figures 9 and 14 it can be observed that these parameters for the individual measurement series are similar. In addition, we used the same two modules M1 and M2, which allowed to obtain similar results. For instance, in Figure 3 the changes in the permeate flux after the membranes stabilized (Lines 2) are similar.

10) In order to organize this section, Figures 4 and 5 should be in one Figure (M1), similar to Figures 7 and 8 (M4), Figures 9 and 10, and figures 12 and 13.

The presentation of both data on the permeate flux (Y1) and turbidity (Y2) in one graph will significantly worsen the readability of a graph. Therefore, we believe that the separation of so many relevant data ensures a much better presentation and interpretation of the results obtained in the present study.

11) Although the objective of this study is to compare the different microfiltration membranes and module configuration, the results are described according to the type of fermentation used. The authors should simplify these sections and give more relevance to the behavior of the membranes after cleaning.

This comment, like the first one, suggests that the article is not well organized. Therefore, as state before, we respectfully disagree with this comment, for the reasons set out in the response to the first comment. In turn, the impact of chemical cleaning on the membranes behaviour (properties and performances) has been extensively discussed, for instance:

-      lines 376-382,

-      lines 418-422,

-      lines 419-420,

-      lines 506-515,

-      lines 549-551.

12) The number of references is too high (112) to be a research article. Authors should select only citations that are relevant in the text.

Thank you for the suggestion. The number of references has been significantly reduced (to 88). 

Reviewer 2 Report

Manuscript ID : Membranes-1052412

The authors have tested the performance of PP and ceramic microfiltration membrane.

Membranes have been used to clarify 1,3-propanediol fermentation broths and suspensions of yeast wastewater. The authors have done considerable experimental work. The work is convincing regarding the methodology. Some points should be considered before possible publication in membranes journal:

  • In table 1, authors have to write the chemical structure correctly such as MgSO4·7H2O should be MgSO7H2O

  • The abbreviation of TMP should be mentioned: Trans Membrane Pressure (TMP)

  • In table 2, the unit of d1 is not clear

  • In figure 2(a), the porosity of the membrane surface is different . The authors have to discuss the effect of gelling bath on the performance?

  • Figure 2 (c) does not show sufficient details on the cross section

  • In lines 172-176, authors have to refer to the difference between M1 and M2

  • Authors have to summarize the common cleaning agents (in table): Pros and cons on the polymeric and ceramic membrane performance

  • Authors have stated that" As reported before, an operation of wetting the PP membranes with ethanol was carried out prior to the MF experiments" . The authors have to cite this paragraph.

  • In figure 3, the authors have to interpret the change in M3 (1) and M3(2).

  • Authors have to highlight the water quality (summary) using the different modules even if they have published it previously. No need to give it in details.

  • In figure 6, authors have to refer to the MF time of the SEM result?

  • Figure 11 may show slight degradation of PP membrane however the authors may need to conduct mechanical testing (tensile test) to ensure the pp degradation.

  • In conclusion, authors have stated " very high efficiency in purifying the biological suspensions have been demonstrated" however the authors did not discuss the water purity in this research.

Author Response

Review 2

The authors have tested the performance of PP and ceramic microfiltration membrane. Membranes have been used to clarify 1,3-propanediol fermentation broths and suspensions of yeast wastewater. The authors have done considerable experimental work. The work is convincing regarding the methodology. Some points should be considered before possible publication in membranes journal:

1)    In table 1, authors have to write the chemical structure correctly such as MgSO4·7H2O should be MgSO7H2O

In the submitted version of the article we have used the subscripts in the nomenclature of chemical formulas. However, the article file downloaded from the Membranes website is without the subscripts, we do not know why. Therefore, we have corrected it. Before printing, we will check the article very carefully and we will successfully eliminate all such errors.

2)    The abbreviation of TMP should be mentioned: Trans Membrane Pressure (TMP)

Thank you for the suggestion. We have corrected it (line 184).

3)    In table 2, the unit of d1 is not clear

The unit of d1 is µ [micrometers], as we wrote in the submitted version of the article. However, similar to the nomenclature of chemical formulas, the article file downloaded from the Membranes website does not present this symbol correctly. Before printing, we will check the article proof very carefully and we will successfully eliminate all such errors.

4)    In figure 2(a), the porosity of the membrane surface is different . The authors have to discuss the effect of gelling bath on the performance?

We agree that the effect of gelling bath on the performance would be interesting to know.  Unfortunately, we do not have the information about gelling bath performed, therefore, we cannot discuss such effect. Moreover, we feel that it falls outside the scope of this study.

Moreover, in each of the experiments performed, cells are much larger than diameters of surface pores. Hence, a filter-cake was quickly formed on the membranes surface and the properties of the -formed dynamic membrane determined the process performance, that can be observed, for example, in the improvement of the permeate NTU value.

5)    Figure 2 (c) does not show sufficient details on the cross section.

Despite a lot of spraying attempts, the material collected charges very strongly and it was not possible to make higher useful magnifications. We have changed the Figure 2c and 2d for the better one.

Although the pores are not clearly visible, it can be observed that on the coarse-grained base there is a thin selective layer.

6)    In lines 172-176, authors have to refer to the difference between M1 and M2.

There is no difference between the modules M1 and M2. The module M1 has been used to separate fermentation broths with Citrobacter freundii bacteria, whereas the module M2 has been used to separate broths with Lactobacillus casei bacteria.

7)    Authors have to summarize the common cleaning agents (in table): Pros and cons on the polymeric and ceramic membrane performance.

A summary of the common agents used for cleaning ceramic membranes fouled by biological suspensions has been presented in our recently published paper:

Tomczak, W.; Gryta, M. Cross-flow microfiltration of glycerol fermentation broths with Citrobacter freundii. Membranes 2020, 10, 67, doi:10.3390/membranes10040067

Therefore, in the present study, such summary has been presented in the form of the following sentence:

Lines 92-95: It is an important issue which must be considered, since for cleaning membranes fouled by bacterial suspensions, the most commonly used are aggressive cleaning agents, such as: sodium hydroxide (NaOH) and sodium hypochlorite (NaOCl) [5].

With regard to polymeric membranes, in the available literature studies investigating their chemical cleaning after MF of the fermentation broths are very limited. Therefore, it is impossible to determine the most commonly used cleaning agents for polymeric membranes fouled by such bacterial suspensions. Nevertheless, based on the literature review, we have demonstrated that chemical cleaning of polymeric membranes may lead to modification of their hydraulic performances, mechanical properties and physical structures (lines 95-105).

Pros and cons of the polymeric and ceramic membrane performance has been widely discussed in the Introduction:

-      Pros of the ceramic membranes: lines 58-67, 74-79,

-      Cons of the ceramic membranes: lines 68-69, 80-84,

-      Pros of the polymeric membranes: lines 88-91,

-      Cons of the polymeric membranes: lines 91-92, 95-105.

8)    Authors have stated that" As reported before, an operation of wetting the PP membranes with ethanol was carried out prior to the MF experiments" . The authors have to cite this paragraph.

Thank you for the suggestion. We have changed this sentence to the following: As reported in Section 2.2, an operation of wetting the PP membranes with ethanol was carried out prior to the MF experiments.

9)    In figure 3, the authors have to interpret the change in M3 (1) and M3(2).

The new membrane, after washing the ethanol with water, shows a strong tendency to push the water out of the pores and regain its hydrophobicity. If the filling is heterogeneous in the pores, (e.g. water-air-water-air system) and the membrane is immersed in ethanol, alcohol quickly floods the air space. On the other hand, it is difficult to obtain a concentrated ethanol solution in flooded by water of pore parts. Obviously, dilute ethanol solution, e.g. 60% is not as easy to press through the pores as it is for a concentrated solution > 90%. Hence, better wetting effects of the membranes were obtained by using significantly higher TMP values for the membrane in the M3 module and hence Lines (1) and (2) are so close to each other in Fig. 3 – Module M3.

We have improved the manuscript by adding the following explanation:

- lines: 235-237: Probably, the new PP membrane, after washing the ethanol with water, shows a strong tendency to push the water out of the pores and part of its volume is again filled by air, which blocks the flow of water through the pores.

10) Authors have to highlight the water quality (summary) using the different modules even if they have published it previously. No need to give it in details.

In this study, the quality of water was not investigated, but the quality of the obtained permeate after MF of fermentation broths. The main purpose of this study was to obtain sterile broth, which can ssubsequently be separated using the NF or MD processes, as it has been presented in the following publications:

-      Bastrzyk J., Gryta M., 2015. Separation of post-fermentation glycerol solution by nanofiltration membrane distillation system. Desalination and Water Treatment 53, 319-329. DOI: 10.1080/19443994.2013.839402

-      Bastrzyk J., Gryta M., Karakulski K., 2014. Fouling of nanofiltration membranes used for separation of fermented glycerol solutions. Chemical Papers 68, 757-765. DOI: 10.2478/s11696-013-0520-8

It is worth noting that results obtained from ion chromatography and HPLC analysis showed that the compositions of feed and permeate were the same. In some papers authors discuss a desalination effect in the MF process (resulting from the formation of a cake layer), but we did not observe this phenomenon in our work.

11) In figure 6, authors have to refer to the MF time of the SEM result?

Thank you for the suggestion. We have referred to the MF time of the SEM result.

Figure 6. SEM image of PP S6/2 membrane (module M1) surface covered by deposit after MF process of fermentation broth with Citrobacter freundii. Deposit formed during 6 h of MF process.

12) Figure 11 may show slight degradation of PP membrane however the authors may need to conduct mechanical testing (tensile test) to ensure the pp degradation.

In the present work, we did not study it and, unfortunately, we cannot do it now. It is due to the fact that the experiments were performed over two years ago, so after that time the membranes underwent further degradation in contact with air. However, it is worth noting that in another study, for the same membranes, due to significant oxidative degradation of membranes, tensile strength decreased from 2.7 to 2.3 MPa and elongation at break from 98 to 78%.

13) In conclusion, authors have stated " very high efficiency in purifying the biological suspensions have been demonstrated" however the authors did not discuss the water purity in this research.

As stated earlier (response to the 10th comment), the quality of the obtained permeate after MF of fermentation broths was examined in this study. Indeed, the purification efficiency of biological suspensions is determined by turbidity removal and sterility of the permeate obtained. Therefore, since the performed MF allowed almost complete removal of the solution turbidity and obtain a sterile permeate, the efficiency of the process was assessed as very high.

Reviewer 3 Report

The article entitled "Comparison of polypropylene and ceramic microfiltration membranes applied for separation of 1,3-PD fermentation broths and Saccharomyces cerevisiae yeast suspensions" written by Tomczak and Gryta has a significant amount of high-quality scientific conclusions with a wide potential for practical application. In the Introduction section a great overview of the characteristics of ceramic membranes as well as the reasons for looking for their alternative are shown. Also, it is clearly explained why 1,3-propanediol (1,3-PD) fermentation broths and suspensions of yeast Saccharomyces cerevisiae were chosen for examination. The discussion of the obtained results is of very high quality, and conclusions are not exclusively scientific but very applicable in practice.

Please see below two minor remarks:

  1. Please give an assessment of the applicability of the results of this work to the MF of other fermentation/cultivation broths.
  2. Lines 250-251: Obviously, membranes with larger pore size are more porous, which means they provide a higher permeate flux. Please explain why polypropylene and ceramic membranes with the same pore sizes were not compared and give an assessment of the behavior of the system in such conditions.
  3. Figure 4: Please explain on the basis of which criteria the moments of water filtration are selected.
  4. Section 3.4: Please evaluate the viability of yeast cells after separation under the applied conditions.

Author Response

Review 3

The article entitled "Comparison of polypropylene and ceramic microfiltration membranes applied for separation of 1,3-PD fermentation broths and Saccharomyces cerevisiae yeast suspensions" written by Tomczak and Gryta has a significant amount of high-quality scientific conclusions with a wide potential for practical application. In the Introduction section a great overview of the characteristics of ceramic membranes as well as the reasons for looking for their alternative are shown. Also, it is clearly explained why 1,3-propanediol (1,3-PD) fermentation broths and suspensions of yeast Saccharomyces cerevisiae were chosen for examination. The discussion of the obtained results is of very high quality, and conclusions are not exclusively scientific but very applicable in practice.

Please see below two minor remarks:

1)    Please give an assessment of the applicability of the results of this work to the MF of other fermentation/cultivation broths.

We believe that the results obtained in the present work may be useful to carry out the MF process of other fermentation/cultivation broths. Indeed, we have presented universal relations, for instance, the fact that the quality of the obtained permeate depends on the membrane pore size or positive impact of the feed flow velocity on the relative permeate flux. However, it should always be taken into account that the fermentation broths are very complex medium, the composition of which strictly depends on the carbon source, bacteria strain, fermentation conditions as well as several other factors. Therefore, the results of our work give an overall view of the MF process efficiency for biological suspensions, however, all the characteristic dependencies should be examined individually.

We have improved the manuscript by adding the following sentences:

-      lines: 609-614: However, it should always be taken into account that the fermentation broths are very complex medium, the composition of which strictly depends on the carbon source, bacteria strain, fermentation conditions as well as several other factors. Therefore, the results of presented work give an overall view of the MF process efficiency for biological suspensions, however, all the characteristic dependencies should be examined individually.

2)    Lines 250-251: Obviously, membranes with larger pore size are more porous, which means they provide a higher permeate flux. Please explain why polypropylene and ceramic membranes with the same pore sizes were not compared and give an assessment of the behavior of the system in such conditions.

Indeed, polypropylene and ceramic membranes with the same pore sizes were not compared. It was due to the fact, that in order to compare the membranes performance, it is necessary to ensure the same system conditions (the same membrane module) since even small changes in the structure of the module channels may have a great influence on flow turbulence and thus on shear forces. The membrane module in the installation used is intended for ceramic membranes, and the Accurel PP V8 / 2 membrane was the only polymeric membrane with parameters similar to those of the ceramic membrane used.

3)    Figure 4: Please explain on the basis of which criteria the moments of water filtration are selected.

The results presented in this paper are part of the great work which has been focused on the 1,3-propanediol production via glycerol fermentation process. The M1 module has been connected with the bioreactor whose operation imposed filtration periods. Indeed, we agree with the Reviewer that this topic could be the subject of another studies, such as determining the optimal running and rinsing periods.

4)    Section 3.4: Please evaluate the viability of yeast cells after separation under the applied conditions.

In the present work, the viability of yeast cells after separation process has not been studied. However, it is worth noting, that in the author's previous publication:

  • Barancewicz, M. Gryta, Ethanol production in a bioreactor with an integrated membrane distillation module, Chemical Papers 66 (2) 85–91 (2012), DOI: 10.2478/s11696-011-0088-0

the same installation has been used as a bioreactor for ethanol production. It has been demonstrated that high flow velocities do not have the impact on the yeast viability. Indeed, a very good productivity, close to the theoretical, was obtained.

Round 2

Reviewer 1 Report

The changes made in the manuscript have improved it.